# Multi-Object Representation Learning via Feature Connectivity and Object-Centric Regularization

**Alex Foo**    **Wynne Hsu**    **Mong Li Lee**
School of Computing
National University of Singapore
{alexfoo,whsu,leeml}@comp.nus.edu.sg

## Abstract

Discovering object-centric representations from images has the potential to greatly improve the robustness, sample efficiency and interpretability of machine learning algorithms. Current works on multi-object images typically follow a generative approach that optimizes for input reconstruction and fail to scale to real-world datasets despite significant increases in model capacity. We address this limitation by proposing a novel method that leverages feature connectivity to cluster neighboring pixels likely to belong to the same object. We further design two object-centric regularization terms to refine object representations in the latent space, enabling our approach to scale to complex real-world images. Experimental results on simulated, real-world, complex texture and common object images demonstrate a substantial improvement in the quality of discovered objects compared to state-of-the-art methods, as well as the sample efficiency and generalizability of our approach. We also show that the discovered object-centric representations can accurately predict key object properties in downstream tasks, highlighting the potential of our method to advance the field of multi-object representation learning.

## 1 Introduction

Human understanding of the world relies on objects as compositional building blocks [24], and emulating this through object-centric representations can improve robustness, sample efficiency, generalization to out-of-domain distributions, and interpretability of machine learning algorithms [18, 9]. Recent work utilizes a generative approach, optimizing pixel-based reconstruction to learn object-centric representations [8, 17, 33, 4, 35, 12, 13, 11, 45, 46, 51]. This approach has limitations as it prioritizes pixel accuracy over object discovery and functional feature extraction [30, 11]. This may lead to the failure of discovering objects [35], or obtaining useful object features, such as position, shape, or boundaries between overlapping objects [23]. Additionally, pixel-based reconstruction tends to waste model capacity on less important visual features, such as complex backgrounds [25], making scaling these methods to real-world images a challenge.

To address the fundamental limitations of pixel-based reconstruction, we propose a framework that leverages feature connectivity and design two object-centric regularization terms to directly refine object representations, ensuring sufficient separation and high disentanglement between dimensions. Our method utilizes visual connectedness principles [38], where similar pixels that are connected should belong to the same object, to guide object discovery. The two regularization terms promote disentangled representations and prevent sub-optimal clustering.

We demonstrate that our approach outperforms state-of-the-art methods in discovering multiple objects from simulated, real-world, complex texture and common object images in a fine-grained manner without supervision. The proposed solution attains sample efficiency and is generalizable to out-of-domain images. The learned object representations also accurately predict key object

37th Conference on Neural Information Processing Systems (NeurIPS 2023).

properties in downstream tasks. Our contributions include: (1) a framework that leverages feature connectivity for fine-grained object discovery, (2) introduction of object-centric regularization terms as an alternative to pixel-based reconstruction loss, (3) experimental validation of our solution's superior performance, and (4) demonstration of the usefulness of discovered object representations in downstream tasks.

## 2   Related Work

Numerous works have demonstrated remarkable success in segmenting real-world images. Most of these works focus on semantic segmentation and object detection by utilizing supervised signals [19, 6]. In contrast, unsupervised approaches like SLIC [1] employ a modified k-means algorithm to cluster pixels into superpixels, similar to Felzenszwalb's algorithm [15] that relies on hand-crafted features for clustering. However, these methods do not focus on learning useful representations for the segmented components.

Various unsupervised methods for learning object-centric representations have been proposed, and can be categorized into three main approaches: spatial attention, sequential attention, and iterative attention. Spatial attention approach utilizes spatial transformer networks [21] to crop out rectangular regions from an image and extract object attributes such as position and scale [14, 31, 8, 33]. They rely on a fixed-size sampling grid which may not be suitable for scenes with varying object sizes, and may compromise training when the sampling grid does not overlap with any object.

The sequential attention approach uses RNN-based models such as MONet [5] and GENESIS [12] to sequentially attend to different regions in an image. These methods employ a deterministic network to perform the attention process, which allows them to capture and represent objects in the scene. However, these methods may neglect smaller objects as they tend to produce a weaker signal during the attention process. This can lead to incomplete or biased representations of scenes with objects of varying sizes. To overcome this, GENESIS-V2 [13] uses a stochastic stick-breaking process to perform attention randomly.

In the iterative attention approach, a set of object representations is randomly initialized and then iteratively refined to bind these objects to different regions of an image. IODINE [17] is a model that can discover objects with disentangled representations. However, it requires long training times and many samples. Slot Attention [35] introduces competition among the object representations by utilizing cross-attention along the object dimension. While this method is fast, versatile and can be extended to handle videos [29], it may fail to discover objects when the training set is diverse, and the resulting representations are highly entangled.

EfficientMORL [11], SLATE [45], SysBinder [46] and BO-QSA [22] are recent developments in the iterative attention approach aimed at addressing some limitations of earlier methods like Slot Attention. EfficientMORL presents an hierarchical variational autoencoder and a lightweight iterative refinement network to increase efficiency without sacrificing representation quality. SLATE increases the non-linear interaction between the slots in Slot Attention with an autoregressive decoder that is conditioned on the slots, resulting in improved reconstructions and object-centric representations. SysBinder enhances the slots of Slot Attention with factor representations called block-slots, which provides within-slot disentanglement between learned factors. BO-QSA initializes the slots of Slot Attention as learnable embeddings instead of sampling from a Gaussian distribution and uses bi-level optimization, resulting in more stable training. Despite the advancements, one drawback remains: the number of clusters are fixed a priori which limits the applicability in real-world scenarios where the number of objects or clusters is not known beforehand.

## 3   Methodology

Our proposed method OC-Net is designed to extract objects in an image without relying on labeled data or specifying the number of objects present in the image. By not requiring the number of objects to be specified beforehand, OC-Net can generalize better to real-world scenes with varying numbers of objects and handle more complex scenarios.

Figure 1 shows the main components of OC-Net. The main idea behind OC-Net is to learn pixel embeddings that can be clustered to discover objects and their respective object masks and object

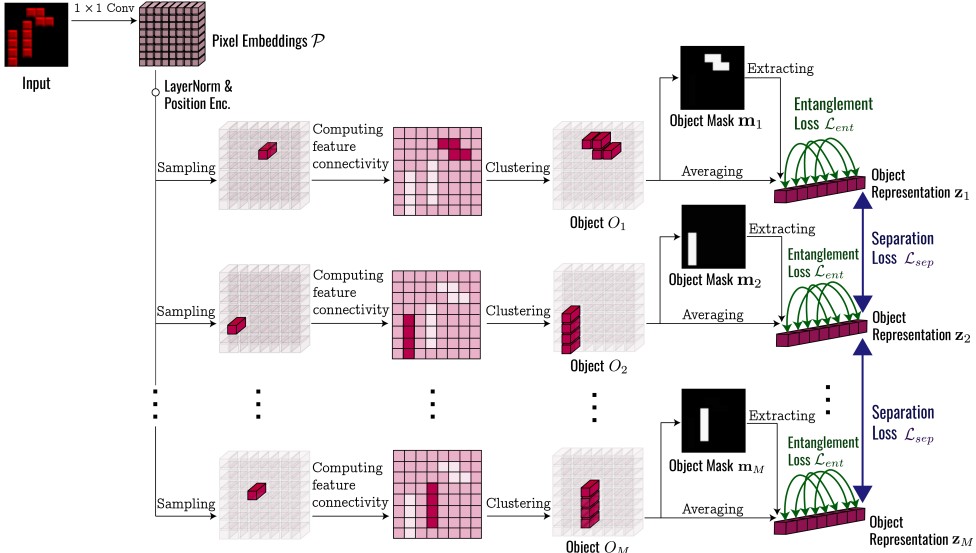

Figure 1: Overview of OC-Net.

representations. This is achieved by passing the input image through a $1 \times 1$ convolutional layer to obtain a set of $N$ pixel embeddings $\mathcal{P} = \{\mathbf{p}_1, \ldots, \mathbf{p}_N\}$ of $D$ dimensions each. We leverage on feature connectivity and iteratively cluster the embeddings of neighbouring pixels based on the likelihood that they belong to the same object. The output is a set of objects $\mathcal{O} = \{O_1, \ldots, O_M\}$ where each object is a set of pixel embeddings. We derive the object mask $\mathbf{m}_j$ of each object by setting the pixel corresponding to the embedding in $O_j$:

$$\mathbf{m}_j[i] = \begin{cases} 1 & \text{if } \mathbf{p}_i \in O_j \\ 0 & \text{otherwise} \end{cases} \tag{1}$$

where $\mathbf{m}_j[i]$ denotes the $i^{th}$ pixel value and $i \in \{1, \ldots, N\}$.

With this, we obtain the matrix of object representations $\mathbf{Z} = [\mathbf{z}_1, \ldots, \mathbf{z}_M]$ where each $\mathbf{z}_j$ is the sum of extracted mask information and the average of the pixel embeddings in $O_j$:

$$\mathbf{z}_j[d] = (\mathbf{A} \cdot \mathbf{m}_j)[d] + \frac{1}{|O_j|} \sum_{\mathbf{p}_i \in O_j} \mathbf{p}_i[d] \tag{2}$$

where $\mathbf{z}_j[d]$ denotes the $d^{th}$ value of vector $\mathbf{z}_j$, $d \in \{1, \ldots, D\}$, $\mathbf{A}$ is the mask transformation matrix.

### 3.1 Object Discovery

The object discovery process iteratively clusters the pixel embeddings based on their feature connectivity and similarity. LayerNorm [2] is applied to normalize all pixel embeddings, and positional encodings are added to the pixel embeddings. The neighbors of a pixel embedding $\mathbf{p}$ are the set of embeddings of the 8 neighbours in the input image. We use Dijkstra's algorithm to compute the shortest distance of a sampled pixel embedding to all other embeddings as follows.

Let $\mathcal{U}$ be the set of pixel embeddings that have not been assigned to an object yet. We uniformly sample a pixel embedding $\mathbf{p}_i \in \mathcal{U}$. The distance from $\mathbf{p}_i$ to itself is set to zero, and the distance to all the other pixels is set to infinity. We select an unvisited pixel embedding $\mathbf{p}_m$ that has the minimum distance to $\mathbf{p}_i$. Let $\mathbf{p}_k$ be the embedding of a neighbour of the pixel corresponding to $\mathbf{p}_m$. We compute the distance between a pair of neighbouring pixels as the similarity between their corresponding embeddings given by:

$$\text{sim}(\mathbf{p}_m, \mathbf{p}_k) = \sqrt{\sum_{d=1}^{D} (\mathbf{p}_m[d] - \mathbf{p}_k[d])^2} \tag{3}$$

where $\mathbf{p}_m[d]$ denotes the $d^{th}$ value of the embedding $\mathbf{p}_m$.

If the distance between $\mathbf{p}_i$ and $\mathbf{p}_k$ is shorter through $\mathbf{p}_m$, we update the shortest distance accordingly. This ensures that we consider the most efficient path between pixel embeddings, leading to better object discovery. We mark $\mathbf{p}_m$ as visited and consider it to be part of the same object as $\mathbf{p}_i$ according to a threshold. The process is repeated for the next unvisited pixel embedding until all pixels have been visited.

## 3.2 Object-Centric Regularization

We design two object-centric regularization terms $\mathcal{L}_{sep}$ and $\mathcal{L}_{ent}$ to improve the quality of the learned object representations for downstream generalization and object discovery. Given the matrix of object representations $\mathbf{Z} = [\mathbf{z}_1, \ldots, \mathbf{z}_M]$ corresponding to the object training samples $\{(\mathbf{x}_i, \mathbf{y}_i)\}_{i=1}^M$, we quantify downstream generalization performance with the prediction error:

$$\delta_{\mathbf{Z}} = \frac{1}{M} \sum_{i=1}^{M} ||\mathbf{W} \cdot \mathbf{z}_i - \mathbf{y}_i|| \tag{4}$$

where $\mathbf{W}$ is the minimum-norm solution of the downstream predictor.

Since labels $\mathbf{y}_i$ are unknown in the unsupervised setting, $\delta_{\mathbf{Z}}$ cannot be directly minimized. Theorem 3.1 below shows that we can minimize an upper bound of $\delta_{\mathbf{Z}}$ by using the projection matrix $\mathbf{P}_{\mathbf{Z}}$. Proof of the theorem is provided in the supplementary material.

**Theorem 3.1.** *Let $\mathbf{Y}$ be the matrix of training sample labels and $\mathbf{P}_{\mathbf{Z}}$ be the projection matrix of $\mathbf{Z}$:*

$$\mathbf{P}_{\mathbf{Z}} = \mathbf{I} - \mathbf{Z}^\top (\Sigma_{\mathbf{Z}})^\dagger \mathbf{Z} \tag{5}$$

*where $\mathbf{I}$ is the identity matrix, $(.)^\dagger$ is the pseudoinverse and $\Sigma_{\mathbf{Z}} = \mathbf{Z}\mathbf{Z}^\top$ is the unnormalized covariance matrix of $\mathbf{Z}$. Let $||.||_F$ be the Frobenius norm. Then, the following relation holds:*

$$\delta_{\mathbf{Z}} \leq ||\mathbf{P}_{\mathbf{Z}}||_F ||\mathbf{Y}||_F \tag{6}$$

Since $||\mathbf{Y}||_F$ in Equation 6 is unknown but fixed, we can minimize $\delta_{\mathbf{Z}}$ by minimizing $||\mathbf{P}_{\mathbf{Z}}||_F$. From the definition of $\mathbf{P}_{\mathbf{Z}}$ in Equation 5, $||\mathbf{P}_{\mathbf{Z}}||_F$ is minimized when the rank of $\Sigma_{\mathbf{Z}}$ is maximized [44, 3]. We achieve this by maximizing the diagonal entries of $\Sigma_{\mathbf{Z}}$ with a separation term $\mathcal{L}_{sep}$ while simultaneously minimizing its off-diagonal entries with an entanglement term $\mathcal{L}_{ent}$, in effect regularizing $\Sigma_{\mathbf{Z}}$ to be a diagonal matrix with a maximum number of nonzero entries.

Maximizing the diagonal entries of $\Sigma_{\mathbf{Z}}$ via $\mathcal{L}_{sep}$ consequently maximizes the distance between object representations in the latent space. This encourages the model to learn distinct and non-overlapping object representations. Expanding the representation space also ensures that objects with varying features and properties can be accurately represented and distinguished from one another, enhancing both downstream generalization and fine-grained object discovery. We define $\mathcal{L}_{sep}$ as:

$$\mathcal{L}_{sep} = \frac{1}{D} \sum_{d=1}^{D} \max(0, 1 - \sqrt{\sigma_d + \tau}) \tag{7}$$

where $\sigma_d$ is the variance of the $d^{th}$ dimension across the vectors $\mathbf{z}_1, \cdots, \mathbf{z}_M$ and $\tau$ is a small constant to maintain numerical stability.

The entanglement term $\mathcal{L}_{ent}$ minimizes the off-diagonal entries of $\Sigma_{\mathbf{Z}}$ and consequently minimizes the correlation between dimensions in the latent space $\mathbf{Z}$, thereby achieves more disentangled object representations. Such representations are easier to manipulate and analyze, as each dimension captures a distinct object property, such as position, scale, or color. $\mathcal{L}_{ent}$ is defined as follows:

$$\mathcal{L}_{ent} = \frac{1}{D \times (M-1)} \sum_{i \neq j} \Sigma_{\mathbf{Z}}[i, j] \tag{8}$$

## 4 Performance Study

We conduct experiments to evaluate the performance of OC-Net in terms of quality, sample efficiency and generalizability. We use a diverse range of datasets to demonstrate its effectiveness across various scenarios:

Table 1: Summary of dataset characteristics

| Dataset | Type | Ground Truth | Image Size | # Samples |
|---------|------|--------------|------------|-----------|
| Multi-dSprites | Simulated | Pixel Mask | $64 \times 64$ | 1M |
| Tetrominoes-NM | Simulated | Pixel Mask | $35 \times 35$ | 1M |
| SVHN | Real-World | Bounding Box | Varied | 530K |
| IDRiD | Real-World | Pixel Mask | $4288 \times 2848$ | 81 |
| CLEVRTEX | Complex Texture | Pixel Mask | $128 \times 128$ | 50K |
| CLEVRTEX-OOD | Complex Texture | Pixel Mask | $128 \times 128$ | 10K |
| Flowers | Common Object | Pixel Mask | $128 \times 128$ | 7K |
| Birds | Common Object | Pixel Mask | $128 \times 128$ | 11K |
| COCO | Common Object | Pixel Mask | $128 \times 128$ | 12K |

1. Simulated datasets Multi-dSprites [23] and Tetrominoes-NM. The former consists of multiple oval, heart or square-shaped sprites with some occlusions, while the latter is a subset of the original Tetrominoes dataset [23] where images whose ground truth segmentation requires knowledge of the object shapes are filtered out.

2. Real-world multi-object datasets SVHN [36] and IDRiD [40]. SVHN consists of street view images of house numbers while IDRiD is the Indian Diabetic Retinopathy Image Segmentation Dataset.

3. Complex texture datasets CLEVRTEX [26] and CLEVRTEX-OOD. CLEVRTEX features scenes with diverse shapes, textures and photo-mapped materials while CLEVRTEX-OOD is the CLEVRTEX out-of-distrbution test set with 25 new materials and 4 new shapes.

4. Common object datasets Flowers [37], Birds [48] and COCO [50]. The Flowers dataset features 17 diverse flower classes with large variations viewpoint, scale, illumination and background. Birds is the most widely-used CUB-200-2011 dataset for fine-grained visual categorization. COCO is the variant of the Microsoft Common Objects in Context dataset used for large-scale object segmentation [32].

Table 1 shows the dataset characteristics. Following [35, 11], we use the first 60K samples in Multi-dSprites, Tetrominoes-NM and SVHN for training and hold out the next 320 samples for testing. For IDRiD, we split this dataset into 54 images for training and 27 images for testing. For CLEVRTEX, we use the first 40K samples for training and last 5K samples for testing. For CLEVRTEX-OOD, we use 10K samples for testing. For Flowers, we use the first 6K samples for training and last 1K samples for testing. For Birds, we use the first 10K samples for training and last 1K samples for testing. For COCO, we use the first 10K samples for training and last 2K samples for testing.

We compare OC-Net with SLIC [1], Felzenszwalb [11], Slot Attention [35], EfficientMORL [11], GENESIS-V2 [13], SLATE [45], SysBinder [46] and BO-QSA [22]. We train OC-Net for 1000 iterations with a batch size of 64 using Adam [28] with a learning rate of $1 \times 10^{-3}$. We carried out an initial experiment to choose the clustering threshold. The results show that the value can range from 0.2 to 2.0 without affecting the performance of OC-Net. As such, we set the threshold to $\epsilon = 0.7$ so that two pixels will belong to the same object if their normalized feature similarity is more than 50%. If a pixel is assigned to multiple objects, we assign it to the mask of the first object in that list and ignore its membership in other objects. Training on 64-by-64 images from Multi-dSprites on a single V100 GPU with 32GB of RAM takes about 10 minutes.

For all methods, we set the maximum number of foreground objects to 6 and 4 for Multi-dSprites and Tetrominoes respectively. Training is carried out for 300,000 iterations with a batch size of 64, using the Adam optimizer with a base learning rate of $4 \times 10^{-4}$. We set the size of the latent space to be $D = 64$ for all models. For SVHN and COCO, the number of objects is set to 6. For IDRiD, the number of objects is set to 20 and we train them for 100,000 iterations. For CLEVRTEX and CLEVRTEX-OOD, the number of objects is set to 11. For Flowers and Birds, the number of objects is set to 2.

We use the Adjusted Rand Index (ARI) to measure the quality of objects discovered [20]. The ARI is a measure of similarity between two data clusterings that takes into account the permutation-invariant nature of the predicted segmentation masks and their corresponding ground-truth masks. We also

use the Dice similarity coefficient, along with the Intersection-over-Union (IoU) between the best matching object masks $X$ and $Y$ as follows:

$$\text{Dice}(X, Y) = \frac{2|X \cap Y|}{|X| + |Y|} \quad ; \quad \text{IoU}(X, Y) = \frac{|X \cap Y|}{|X \cup Y|} \tag{9}$$

where $X$ the set of object pixels extracted and $Y$ is the set of annotated object pixels in the ground truth. We compute the mean Dice and the mean IoU scores, denoted as mDice and mIoU respectively, by averaging the individual Dice and IoU scores across all matches. For the complex textures dataset, background discovery is included in the computation of the scores.

## 4.1 Experiments on Quality of Discovered Objects

We first evaluate the ability of OC-Net to discover objects from images with multiple objects. Table 2(a) shows the average ARI, mDice and mIoU scores based on the discovered foreground objects in the simulated datasets after 3 runs. We observe that OC-Net outperforms all other methods by a large margin in Multi-dSprites, and even achieves perfect score for all Tetrominoes-NM test samples.

Table 2(b) shows the results on the real-world multi-object image datasets. For SVHN, OC-Net outperforms all methods, even when the ground truth is provided in the form of bounding boxes. This implies that we need to expand the discovered object masks into their corresponding bounding boxes which are often rough fits, and is the reason for the close difference in mDice and mIoU scores between OC-Net and BO-QSA. For IDRiD, which contains multiple small objects, OC-Net significantly improves the ARI scores and more than doubles the mDice and mIoU scores over all methods, demonstrating its robustness in challenging object discovery tasks.

Table 2(c) shows the results on CLEVRTEX and CLEVRTEX-OOD, which contains complex textured objects and backgrounds. Here, OC-Net again shows superior performance in all metrics, illustrating its capability to effectively segment complex objects. Although a general decrease in performance is observed across all methods in the CLEVRTEX-OOD test set, likely due to a change in data distribution, OC-Net's performance drop is slight and it still outperforms the closest baseline.

Finally, Table 2(d) shows the results on Flowers, Birds and COCO common object datasets. Here, OC-Net again shows superior performance in all metrics, illustrating its capability to effectively segment commonly seen natural objects. Notably, OC-Net outperforms all other methods by a large margin in COCO, demonstrating its robustness in handling objects with highly varied appearances.

Figure 2 visualizes the objects discovered by the various methods for sample images. OC-Net is able to identify large and small objects in Multi-dSprites even when these objects are significantly occluded. Moreover, in the Tetrominoes-NM dataset, despite the presence of shadow effects that often confuse existing methods, OC-Net still manages to separate each tile. For SVHN, only OC-Net is able to segment the character objects out in a fine-grained manner. EfficientMORL tend to group all the characters together while the other methods segment the objects in a coarse-grained manner. For IDRiD, OC-Net is able to segment out the optic disc and small lesions which other methods fail to discover. For CLEVRTEX and CLEVRTEX-OOD, OC-Net is able to segment out the various objects from the complex-textured backgrounds in a fine-grained manner. Finally, for Flowers, Birds, and COCO, only OC-Net is able to segment out the complex-shaped and multi-part objects from the backgrounds in a fine-grained manner.

## 4.2 Experiments on Sample Efficiency

One obstacle to unsupervised object discovery is the availability of a sufficiently large number of suitable training samples. Sample efficiency refers to a model's ability to learn effectively from a relatively small number of examples. Figure 3 shows the mIoU scores as we decrease the number of training samples in Multi-dSprites, SVHN and CLEVRTEX. OC-Net is able to achieve near-optimal performance even with a significantly smaller training set (1,000 samples) compared to all the other methods. The high sample efficiency of OC-Net reduces the need for large, potentially costly or difficult-to-obtain datasets. This makes OC-Net a more practical solution for real-world applications.

Table 2: Evaluation scores for the discovered foreground objects.

(a) Simulated datasets

| Method | Multi-dSprites | | | Tetrominoes-NM | | |
|---|---|---|---|---|---|---|
| | ARI | mDice | mIoU | ARI | mDice | mIoU |
| SLIC | 67.9±0.0 | 78.5±0.0 | 70.8±0.0 | 53.0±0.0 | 66.1±0.0 | 53.6±0.0 |
| Felzenszwalb | 97.4±0.0 | 98.6±0.0 | 95.0±0.0 | 95.0±0.0 | 98.0±0.0 | 96.9±0.0 |
| Slot Attention | 91.3±0.3 | 45.7±0.7 | 32.6±0.6 | 99.8±0.1 | 41.5±0.8 | 26.6±0.7 |
| EfficientMORL | 85.2±0.5 | 30.1±1.3 | 19.5±1.1 | 99.0±1.7 | 42.5±2.3 | 27.6±1.9 |
| GENESIS-V2 | 85.0±1.3 | 81.5±1.9 | 72.2±1.4 | 97.6±0.5 | 47.1±1.1 | 31.0±0.8 |
| SLATE | 89.5±1.2 | 82.5±0.9 | 72.6±1.1 | 84.5±1.5 | 57.8±0.9 | 44.3±0.8 |
| SysBinder | 72.3±1.2 | 30.6±1.1 | 19.6±1.0 | 90.7±1.7 | 41.8±1.9 | 27.0±1.7 |
| BO-QSA | 90.4±1.1 | 91.6±1.1 | 88.0±1.2 | 99.3±0.3 | 40.9±1.4 | 25.8±1.2 |
| OC-Net | **99.8±0.0** | **99.5±0.0** | **99.1±0.0** | **100.0±0.0** | **100.0±0.0** | **100.0±0.0** |

(b) Real-world datasets

| Method | SVHN | | | IDRiD | | |
|---|---|---|---|---|---|---|
| | ARI | mDice | mIoU | ARI | mDice | mIoU |
| SLIC | 5.3±0.0 | 50.1±0.0 | 34.5±0.0 | 32.2±0.0 | 12.7±0.0 | 8.8±0.0 |
| Felzenszwalb | 31.7±0.0 | 51.6±0.0 | 39.8±0.0 | 14.7±0.0 | 19.0±0.0 | 15.4±0.0 |
| Slot Attention | 38.9±1.5 | 51.7±1.8 | 36.7±1.7 | 28.7±1.1 | 8.6±1.7 | 5.0±1.6 |
| EfficientMORL | 32.2±1.7 | 49.2±2.0 | 34.0±1.8 | 16.8±1.5 | 11.1±2.7 | 7.0±1.8 |
| GENESIS-V2 | 28.6±1.4 | 60.8±1.5 | 45.9±1.4 | 18.3±1.6 | 8.8±1.9 | 5.4±1.6 |
| SLATE | 21.2±1.2 | 57.0±1.3 | 41.7±1.5 | 35.6±2.1 | 8.1±1.2 | 4.7±1.8 |
| SysBinder | 15.8±1.6 | 49.5±1.9 | 34.1±1.8 | 25.2±1.3 | 16.6±1.7 | 11.1±1.8 |
| BO-QSA | 24.3±1.2 | 62.0±1.6 | 48.3±1.3 | 27.7±2.0 | 7.0±1.9 | 4.5±1.7 |
| OC-Net | **39.7±0.1** | **64.6±0.1** | **49.9±0.1** | **39.0±0.4** | **38.1±0.2** | **31.2±0.2** |

(c) Complex textures dataset

| Method | CLEVRTEX | | | CLEVRTEX-OOD | | |
|---|---|---|---|---|---|---|
| | ARI | mDice | mIoU | ARI | mDice | mIoU |
| SLIC | 27.4±0.0 | 20.0±0.0 | 13.0±0.0 | 25.8±0.0 | 21.7±0.0 | 14.0±0.0 |
| Felzenszwalb | 57.3±0.0 | 33.6±0.0 | 26.8±0.0 | 44.6±0.0 | 29.6±0.0 | 23.4±0.0 |
| Slot Attention | 58.6±1.6 | 35.0±1.6 | 26.7±1.5 | 51.3±1.9 | 34.1±1.4 | 25.1±1.3 |
| EfficientMORL | 59.5±1.7 | 37.7±1.5 | 31.1±1.4 | 53.9±2.5 | 32.2±2.4 | 25.3±2.8 |
| GENESIS-V2 | 65.6±1.8 | 36.9±1.4 | 30.4±1.4 | 67.6±1.6 | 34.2±1.5 | 27.6±1.9 |
| SLATE | 57.5±1.8 | 33.3±1.6 | 24.4±1.5 | 56.6±1.3 | 34.7±2.1 | 25.3±1.8 |
| SysBinder | 61.4±1.7 | 31.3±1.5 | 23.1±1.4 | 61.0±2.4 | 32.3±2.0 | 23.8±1.8 |
| BO-QSA | **70.9±1.9** | 42.9±1.8 | 34.7±1.7 | 66.1±1.3 | 42.8±1.4 | 33.9±1.3 |
| OC-Net | **70.7±0.9** | **45.1±0.9** | **37.5±0.7** | **69.8±0.8** | **43.5±0.7** | **35.0±0.6** |

(d) Common objects datasets

| Method | Flowers | | Birds | | COCO | |
|---|---|---|---|---|---|---|
| | Dice | IoU | Dice | IoU | mDice | mIoU |
| SLIC | 30.5±0.0 | 18.4±0.0 | 33.1±0.0 | 20.3±0.0 | 36.2±0.0 | 24.4±0.0 |
| Felzenszwalb | 43.7±0.0 | 30.4±0.0 | 34.3±0.0 | 23.0±0.0 | 36.6±0.0 | 27.1±0.0 |
| Slot Attention | 43.0±1.5 | 28.6±1.2 | 42.9±2.0 | 27.9±1.8 | 24.8±2.0 | 15.0±1.7 |
| EfficientMORL | 59.5±2.1 | 45.2±2.2 | 44.0±1.9 | 30.8±1.8 | 28.4±2.3 | 18.9±2.1 |
| GENESIS-V2 | 63.7±2.2 | 50.2±2.2 | 41.4±1.7 | 27.7±1.5 | 25.1±2.1 | 16.1±1.7 |
| SLATE | 55.6±1.2 | 40.8±1.8 | 39.5±1.5 | 25.9±1.8 | 37.0±1.9 | 24.4±1.8 |
| SysBinder | 45.0±1.8 | 30.8±1.6 | 33.7±1.3 | 21.1±2.0 | 18.4±1.6 | 10.7±1.4 |
| BO-QSA | 65.8±1.9 | 51.7±1.9 | 44.6±1.7 | 30.3±1.5 | 34.9±1.1 | 23.6±0.9 |
| OC-Net | **67.2±0.2** | **54.4±0.2** | **47.8±0.2** | **33.5±0.2** | **48.2±0.2** | **35.6±0.2** |

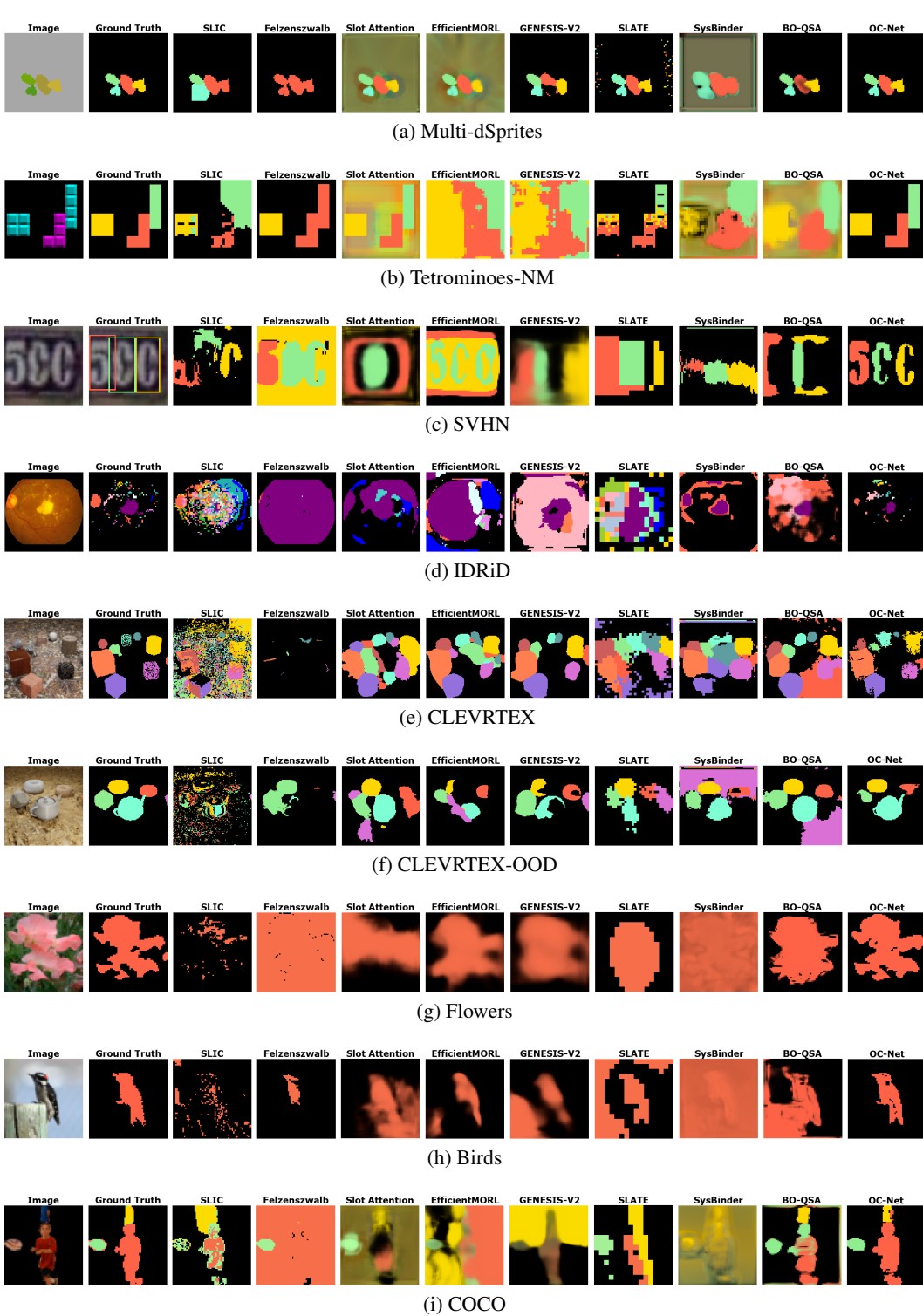

Figure 2: Visualization of discovered objects.

## 4.3 Experiments on Model Generalizability

Ideally, an unsupervised object discovery model should be trained to understand common visual appearances such as the difference between foreground vs background, so as to discover objects from

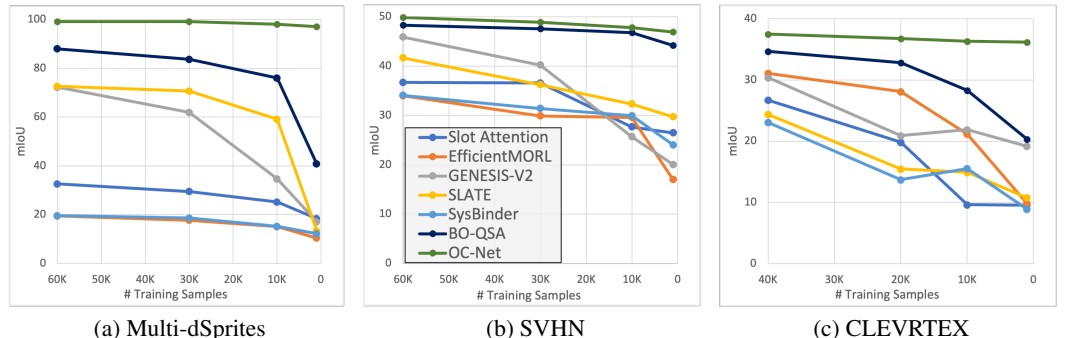

Figure 3: mIoU scores vs decreasing number of training samples.

Table 3: mIoU scores for model generalizability after training on Multi-dSprites.

| Method | Tetrominoes-NM | SVHN | IDRiD | CLEVRTEX | CLEVRTEX-OOD |
|---|---|---|---|---|---|
| Slot Attention | 21.8±3.5 | 19.5±3.8 | 7.5±2.5 | 12.2±2.2 | 12.3±2.2 |
| EfficientMORL | 21.2±3.8 | 23.4±2.8 | 6.5±2.6 | 12.7±3.2 | 15.2±2.0 |
| GENESIS-V2 | 42.9±4.9 | 31.1±2.8 | 8.5±2.4 | 21.9±1.6 | 21.3±2.5 |
| SLATE | 51.4±1.6 | 21.1±2.0 | 10.0±1.7 | 12.7±2.2 | 12.9±1.8 |
| SysBinder | 28.5±1.8 | 23.8±1.1 | 13.9±1.8 | 10.6±1.5 | 11.6±1.9 |
| BO-QSA | 41.8±1.8 | 24.3±2.0 | 4.0±1.5 | 24.4±1.4 | 22.8±2.5 |
| OC-Net | **100.0±0.0** | **47.5±0.5** | **29.1±0.5** | **31.7±0.6** | **31.3±0.6** |

out-of-domain images. In this set of experiments, we compare the generalization ability of OC-Net with the baselines by training the models on Multi-dSprites and testing them on the other datasets.

Table 3 shows the results. For Tetrominoes-NM, there is a decrease in performance for all methods while OC-Net still obtains perfect score. For SVHN, CLEVRTEX and CLEVRTEX-OOD, the performance of all models decrease due to the shift in data distribution. However, OC-Net experiences the smallest drop in performance and still significantly surpasses the best performing method. For IDRiD, the methods show improvement in performance. One possible reason is that the larger training set in Multi-dSprites enables the circular shape of the optic disc to be better segmented. Despite this, OC-Net remains the top performer.

## 4.4 Ablation Studies

Next, we examine the effect of feature connectivity and regularization terms $\mathcal{L}_{sep}$ and $\mathcal{L}_{ent}$ on the performance of OC-Net. We implemented three variants of OC-Net: (a) w/o connectivity. Here, we do not require that a path should exist between $i$ and $k$ when computing dist$[i, k]$; (b) w/o $\mathcal{L}_{ent}$. This network is trained without the entanglement regularization term; (c) w/o $\mathcal{L}_{sep}$. The separation regularization term is not used in the training of OC-Net.

Table 4 shows the mIoU scores for all the datasets. Without feature connectivity, we observe a drop in performance across all datasets since it is common for images to have multiple identical objects, that is, same color and shape. As such, OC-Net w/o connectivity tend to cluster these blocks as a single object.

Removing the entanglement term (OC-Net w/o $\mathcal{L}_{ent}$) leads to a slight decrease in the performance as the object representations may still be separated even when the dimensions are entangled. The largest performance drop in all datasets is seen when the object representation separation term is removed (OC-Net w/o $\mathcal{L}_{sep}$), indicating the importance of having a well-separated object representation space for effective object discovery.

Table 4: mIoU scores for variants of OC-Net.

| Method | Multi-dSprites | Tetro-NM | SVHN | IDRiD | CTEX | CTEX-OOD |
|---|---|---|---|---|---|---|
| OC-Net | **99.1±0.0** | **100.0±0.0** | **49.9±0.1** | **31.2±0.2** | **37.5±0.7** | **35.0±0.6** |
| w/o connectivity | 98.8±0.1 | 89.2±0.1 | 36.6±0.1 | 17.3±0.1 | 20.3±0.9 | 17.7±0.9 |
| w/o $\mathcal{L}_{ent}$ | 98.7±0.3 | 99.6±0.1 | 48.8±0.1 | 25.8±0.2 | 26.4±0.8 | 18.9±0.4 |
| w/o $\mathcal{L}_{sep}$ | 49.3±0.3 | 51.2±0.2 | 35.0±0.2 | 4.0±0.1 | 18.5±0.6 | 11.0±0.4 |

## 5 Prediction based on Learned Object Representation

One characteristic of an effective object-centric representation is its ability to encode object properties such as color, position and shape [42]. In this section, we show that the learned object representations are disentangled and can be used to predict the values of these properties.

Given the representations and their corresponding ground truth values of a target object property, we employ them as features to train a gradient boosted tree (GBT) [16]. To evaluate how well the properties of unseen objects are predicted by the GBT, we use the coefficient of determination $R^2$ [49]. We perform an experiment using the learned object representations from the simulated datasets to predict the properties of object such as color, position and shape. We use the mIoU score to match the discovered object to the ground truth object. We split the 320 test images equally into two sets, one for training the GBT model for each object property, and the other for evaluation.

Table 5 shows the average $R^2$ scores of the GBT models on the evaluation set. The GBT models trained with OC-Net representations achieved the highest $R^2$ scores compared to the models trained using the representations from other methods. This suggests that the object representations learned by OC-Net are effective in encoding the object properties.

Table 5: $R^2$ scores for object property prediction on simulated datasets

| Method | Multi-dSprites | | | Tetrominoes-NM | | |
|---|---|---|---|---|---|---|
| | Color | Position | Shape | Color | Position | Shape |
| Slot Attention | 72.2±12 | 96.8±0.1 | 38.2±0.0 | 86.5±6.5 | 98.7±0.6 | 36.3±0.0 |
| EfficientMORL | 86.5±6.2 | 95.8±0.1 | 61.7±0.0 | 94.9±3.2 | 97.9±0.7 | 68.5±0.0 |
| GENESIS-V2 | 78.1±7.5 | 97.1±0.7 | 75.8±0.0 | 88.1±5.8 | 94.6±2.6 | 37.9±0.0 |
| SLATE | 87.5±0.7 | 90.6±4.4 | 31.7±0.0 | 85.5±3.9 | 89.6±0.7 | 10.5±0.0 |
| SysBinder | 73.6±1.0 | 69.3±3.4 | 33.3±0.0 | 97.9±0.6 | 77.8±2.7 | 19.9±0.0 |
| BO-QSA | 96.3±1.6 | 97.4±0.1 | 75.2±0.0 | 98.1±0.7 | 98.9±0.2 | 52.5±0.0 |
| OC-Net | **98.0±0.6** | **98.3±0.1** | **78.1±0.0** | **100.0±0.0** | **99.4±0.1** | **98.7±0.0** |

## 6 Conclusion

In this work, we have described a framework called OC-Net that learns object-centric representations in a fine-grained manner without supervision. OC-Net leverages on feature connectivity and two new regularization terms to learn disentangled representations and to ensure the representations of different objects are well-separated. From the results of experiments conducted on simulated, real-world, complex texture and common object images, we have demonstrated the superior quality of the object representations over current state-of-the-art. Moreover, we have highlighted the sample efficiency and generalizability of OC-Net. Finally, we have shown how the discovered object representations can be used to predict object properties in a downstream task, indicating its potential use for other computer vision applications where samples and ground truth labels are limited.

There are still obstacles that have to be overcome for successful application of our framework to the full visual complexity of the real world. A natural next step would be to extend OC-Net to handle real-world scenes containing objects with more complex part-whole hierarchies. It is also promising to explore explicit representation of the discovered objects into a dictionary of prototypes to better handle occlusion between objects. Lastly, real-world scenes with multiple objects is still of higher visual complexity than the datasets considered here and reliably bridging this gap is an open problem.

## Acknowledgments and Disclosure of Funding

This research is supported by the National Research Foundation Singapore under its AI Singapore Programme (Award Number: AISG-GC-2019-001-2A).

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
