# OpenReview forum: "Multi-Object Representation Learning via Feature Connectivity and Object-Centric Regularization"
_NeurIPS.cc/2023/Conference — NeurIPS 2023 spotlight_

### Official Review · Reviewer_2W4b · 2023-07-02

**Soundness:** 4 excellent
**Presentation:** 4 excellent
**Contribution:** 4 excellent
**Rating:** 8
**Confidence:** 4

**Summary:**

This paper presents a novel approach towards an unsupervised object-centric representation learning framework. Diverging from previous methodologies that prioritize an input reconstruction scheme, this method emphasizes feature connectivity to cluster neighboring pixels, employing two object-centric regularization losses. The underlying insight for these two losses is the learning of a distinct feature of each object while repelling the object feature between objects by constraining the covariance matrix of the predicted label. To validate their approach, the authors execute their experiments on a variety of image types, ranging from simulated and real-world images to those with complex textures. The results reveal a marked performance improvement when compared to alternative methods, demonstrating the efficacy of the proposed framework.

**Strengths:**

1. The innovative approach proposed in this paper provides food for thought. It suggests that the customary input reconstruction paradigm utilized in prior methods can be substituted with the two object-centric regularization terms introduced in this study. The basic concept underpinning these two loss functions is simple, yet it proves to be highly efficacious.

2. The experimental outcomes presented in the paper are encouraging. The proposed framework demonstrates significant improvement when compared with baseline methods. Moreover, the authors effectively showcase the applicability of the proposed design even when the scale of the dataset diminishes. An ablation study further underscores the relevance and importance of the proposed design in this context.

**Weaknesses:**

1. The scalability of this method to real-world scenes of higher complexity remains a point of ambiguity. Further testing and discussion are necessary to ascertain the effectiveness of this approach in more complex, real-world scenarios.
2. Here is one typo: Line 113, to be.

**Questions:**

1. Beyond the L2 distance employed for feature embeddings, exploring the impact of other choices for feature embeddings on model performance and threshold selection could be beneficial. Various methods such as cosine similarity or Mahalanobis distance, among others, could potentially impact the model performance and threshold choice differently.

2. The mask transformation matrix A's initialization strategy is an area that requires further investigation. An exploration into whether a more intricate network structure could enhance the performance would provide more insight into the system's capabilities and potential limitations. This exploration could open up new possibilities for performance optimization.

**Limitations:**

See weaknesses and questions.

---

> ### Author Rebuttal · Authors · 2023-08-08
>
> Thank you for your detailed review and for the encouraging comments. We provide some responses to the questions as follows:
>
> **Q1: The scalability of this method to real-world scenes of higher complexity remains a point of ambiguity. Further testing and discussion are necessary to ascertain the effectiveness of this approach in more complex, real-world scenarios.**
>
> We have conducted new experiments for further evaluation on additional Flowers, Birds, COCO [4a] datasets which contain objects that have more diverse colors and shapes. We have also added two more recent baselines BO-QSA [21] and SLASH [4b] for comparison as suggested. The results in the table below shows that OC-Net is robust and remains the top performer:
>
> |  | Flowers | Flowers | Birds | Birds | COCO | COCO |
> |---|---|---|---|---|---|---|
> | Metric | Dice | IoU | Dice| IoU | mDice| mIoU |
> | SLASH | 39.91 $\pm$1.48 | 25.98 $\pm$1.15 | 43.45$\pm$1.10 | 28.24$\pm$1.08 | 22.34$\pm$1.10 | 13.37$\pm$1.07 |
> | BO-QSA | 65.77 $\pm$1.90 | 51.70 $\pm$1.93 | 44.61$\pm$1.65 | 30.29$\pm$1.50 | 34.90$\pm$1.14 | 23.57$\pm$0.94 |
> | OC-Net | **67.21 $\pm$0.21** | **54.42 $\pm$0.23** | **47.80$\pm$0.19** | **33.50$\pm$0.17** | **48.18$\pm$0.17** | **35.58$\pm$0.15** |
>
> The objects discovered for the various methods on sample images can be viewed in the PDF attached in our general response to all reviewers. We see that OC-Net is able to segment out the objects in a fine-grained manner.
>
> **Q2: Beyond the L2 distance employed for feature embeddings, exploring the impact of other choices for feature embeddings on model performance and threshold selection could be beneficial. Various methods such as cosine similarity or Mahalanobis distance, among others, could potentially impact the model performance and threshold choice differently.**
>
> We will include these comparisons in the camera ready paper.
>
> **Q3: The mask transformation matrix A's initialization strategy is an area that requires further investigation. An exploration into whether a more intricate network structure could enhance the performance would provide more insight into the system's capabilities and potential limitations. This exploration could open up new possibilities for performance optimization.**
>
> We will include additional description and further studies for the mask transformation matrix in the camera ready paper.
>
> [4a] Yang, Yafei, and Bo Yang. "Promising or elusive? unsupervised object segmentation from real-world single images.", NeurIPS 2022.
>
> [4b] Kim, Jinwoo, et al., "Shepherding Slots to Objects: Towards Stable and Robust Object-Centric Learning", CVPR 2023.

---

### Official Review · Reviewer_T4Hn · 2023-07-04

**Soundness:** 2 fair
**Presentation:** 3 good
**Contribution:** 3 good
**Rating:** 6
**Confidence:** 5

**Summary:**

The paper presents a novel approach to object-centric (multi-object) representation learning. In contrast to conventional sequential or iterative attention-based methods, the proposed method employs a pixel-wise shortest distance-based clustering technique. Moreover, the paper emphasizes the use of regularization-based representation learning as an alternative to traditional reconstruction-based approaches. The inclusion of theoretical proofs in the appendix, combined with extensive empirical studies, serves to substantiate the paper's claims and highlight the superior and robust performance achieved by the proposed methodology.

**Strengths:**

1. The paper is well-written with clear structures, logical flow, and informative figures that effectively convey the research findings.
1. The proposed method introduces novel concepts, such as feature connectivity and regularization-based training losses, which lead to impressive results, not only just high score but also in terms of the sample efficiency, on six diverse datasets. These datasets encompass synthetic (simulated), real-world, and complex textures, and the evaluation covers two fundamental tasks in object-centric learning: object discovery and property prediction.
1. The paper offers theoretical support through a mathematical proof presented in the appendix. This proof establishes an upper bound for the downstream generalization error, enhancing our understanding of the proposed method's performance and its capacity to generalize effectively in downstream tasks.

**Weaknesses:**

1. The primary concern raised by the reviewer revolves around the scalability of the proposed method to other datasets.
   - The reviewer questions the difference between using the original RGB input image (with proper normalization) and the feature map generated by the single 1x1 convolution layer where the single 1x1 convolution layer will act as an injective function from a pixel to a representation vector. It is important to address this aspect and provide a clear distinction between the two approaches.
   - The minimal receptive field (1x1) of the encoder (a single-layer 1x1 convolution) limits the ability to consider local or global context during clustering. This issue is not adequately resolved by the empirical studies since the datasets used lack objects with diverse colors or shapes.
   - The reviewer acknowledges that the positional encoding helps to compensate for this limitation. However, there is a concern about the naive utilization of positional encoding and its ability to capture complex object shapes. Additionally, there is a worry that the resulting clusters might primarily focus on object colors without incorporating positional information.
   - In contrast to the reviewer's concerns, the authors claim in section G.1 that the 1x1 convolution contributes to fine-grained object-centric learning. This conflict needs to be addressed and supported by the authors' analysis and empirical evidence.
   - The reviewer suggests conducting empirical studies on additional datasets with more complex object shapes and color combinations, such as PTR [1], MSN [2], and MOVi [3], as demonstrated in previous works [4,5].
   - Additionally, considering the results obtained by other encoders used in previous methods, such as Slot Attention [6], would further support the authors' claims.


2. Another concern pertains to the rigidity of the feature aggregation process due to the use of a hard clustering algorithm. This rigidity may hinder the proposed method's ability to perform well on downstream tasks and more complex real-world datasets.

[1] Hong, Yining, et al., "PTR: A Benchmark for Part-based Conceptual, Relational, and Physical Reasoning", NeurIPS 2021.

[2] Stelzner, Karl, et al., "Decomposing 3D Scenes into Objects via Unsupervised Volume Segmentation", ICLR 2022.

[3] Greff, Klaus, et al., "Kubric: A Scalable Dataset Generator", CVPR 2022.

[4] Biza, Ondrej, et al., "Invariant Slot Attention: Object Discovery with Slot-Centric Reference Frames", ICML 2023.

[5] Kim, Jinwoo, et al., "Shepherding Slots to Objects: Towards Stable and Robust Object-Centric Learning", CVPR 2023.

[6] Locatello, Francesco, et al., "Object-Centric Learning with Slot Attention", NeurIPS 2020.

**Questions:**

1. In terms of the training schedule, it is necessary to provide a clear description regarding any distinction between "steps" (L157) and "iterations" (L165). This will help readers better understand the experimental details and avoid any potential confusion.
1. In the figures depicting qualitative results (Figure 2, 6-13), there seems to be a discrepancy in how the authors classify backgrounds and visualize them as black. While the majority of backgrounds are depicted as black, there are cases where backgrounds are shown in different colors. This raises a question about whether there is any meaningful distinction between black-colored backgrounds and those with different colors. Clarification is needed to understand the criteria used for classifying and representing backgrounds in the visualizations.
1. The paper utilizes a simplistic approach for positional encoding, which differs from other possible options such as the soft positional embedding used in Slot Attention [1] or fixed sinusoidal positional encoding. Since the chosen method deviates from the conventional approaches, it is important to provide a clear explanation regarding the design and rationale behind the positional encoding used in the paper. This will help readers understand the unique choices made and the implications of the selected approach for the overall methodology.


[1] Locatello, Francesco, et al., "Object-Centric Learning with Slot Attention", NeurIPS 2020.

**Limitations:**

1. To facilitate a comprehensive and fair comparison, it is crucial to include an analysis of the model's speed in the paper. Given that the proposed method involves iterative updates of pixel distances, which may pose challenges for parallelization, there could potentially be a significant speed sacrifice compared to other methods. However, the paper currently lacks any analysis or discussion regarding the speed performance of the proposed method.
1. To enhance the persuasiveness of the paper, it would be beneficial to include some more experiments, such as a) recent studies such as ISA [1] and SLASH [2] for the object discovery task, and b) property prediction results over the complex datasets such as CLEVRTEX, in the final version.
1. The explanation regarding the mask transformation matrix (denoted as A) lacks clarity. It is crucial to provide additional details regarding the initialization and training of the matrix, going beyond the information provided in section A.4. Furthermore, the purpose and necessity of the matrix should be clearly articulated, supported by a thorough description of the procedural intricacies and empirical studies demonstrating its effectiveness.


[1] Biza, Ondrej, et al., "Invariant Slot Attention: Object Discovery with Slot-Centric Reference Frames", ICML 2023.

[2] Kim, Jinwoo, et al., "Shepherding Slots to Objects: Towards Stable and Robust Object-Centric Learning", CVPR 2023.

---

> ### Author Rebuttal · Authors · 2023-08-09
>
> Thank you for your detailed review and for the insightful comments especially on scalability. We provide some responses to the questions as follows:
>
> **Q1: The reviewer questions the difference between using the original RGB input image (with proper normalization) and the feature map generated by the single 1x1 convolution layer**
>
> Using RGB values would treat each image on its own, while applying a convolutional layer enables the method to learn appropriate feature maps based on a dataset of images in order to optimize for downstream generalization.
>
> **Q2: The minimal receptive field of the encoder limits the ability to consider local or global context during clustering. This issue is not adequately resolved by the empirical studies since the datasets used lack objects with diverse colors or shapes. The reviewer suggests conducting empirical studies on additional datasets with more complex object shapes and color combinations such as PTR, MSN, and MOVi, as demonstrated in previous works.**
>
> We have conducted new experiments to further evaluation on additional Flowers, Birds, COCO [3a] datasets which contain objects that are more complex or comparable to PTR, MSN and MOVi. We have also added two more recent baselines BO-QSA [21] and SLASH for comparison as suggested. The results in the table below shows that OC-Net is robust and remains the top performer:
>
> |  | Flowers | Flowers | Birds | Birds | COCO | COCO |
> |---|---|---|---|---|---|---|
> | Metric | Dice | IoU | Dice| IoU | mDice| mIoU |
> | SLASH | 39.91 $\pm$1.48 | 25.98 $\pm$1.15 | 43.45$\pm$1.10 | 28.24$\pm$1.08 | 22.34$\pm$1.10 | 13.37$\pm$1.07 |
> | BO-QSA | 65.77 $\pm$1.90 | 51.70 $\pm$1.93 | 44.61$\pm$1.65 | 30.29$\pm$1.50 | 34.90$\pm$1.14 | 23.57$\pm$0.94 |
> | OC-Net | **67.21 $\pm$0.21** | **54.42 $\pm$0.23** | **47.80$\pm$0.19** | **33.50$\pm$0.17** | **48.18$\pm$0.17** | **35.58$\pm$0.15** |
>
> The objects discovered for the various methods on sample images can be viewed in the PDF attached in our general response to all reviewers. We see that OC-Net is able to segment out the objects in a fine-grained manner.
>
> **Q3: Another concern pertains to the rigidity of the feature aggregation process due to the use of a hard clustering algorithm. This rigidity may hinder the proposed method's ability to perform well on downstream tasks and more complex real-world datasets.**
>
> Our proposed method can be modified to use soft clustering by simply changing the inclusion criteria (Algorithm 1, line 21) and the terminating condition of the object discovery process (Algorithm 1, line 26).
>
> **Q4: In terms of the training schedule, it is necessary to provide a clear description regarding any distinction between "steps" and "iterations".**
>
> We will change "steps" to "iterations" to avoid confusion.
>
> **Q5: In the figures depicting qualitative results (Figure 2, 6-13), there seems to be a discrepancy in how the authors classify backgrounds and visualize them as black.**
>
> Following past work such as Slot Attention [3b], we assign the best-matching regions produced by the various methods as the same color as the objects in the ground truths. Our method does not separately model backgrounds and treats all background of different appearances the same.
>
> **Q6: The paper utilizes a simplistic approach for positional encoding, which differs from other possible options such as the soft positional embedding used in Slot Attention or fixed sinusoidal positional encoding.**
>
> We found that using linear positional encodings help attain better performance especially for downstream property prediction. We have conducted additional experiments to compare alternative positional encodings as suggested and show the results on Multi-dSprites below:
>
> | Property | Color | Position | Shape |
> |---|---|---|---|
> | Sinusoidal | 97.8$\pm$0.8 | 75.2$\pm$4.8 | 43.7$\pm$0.0 |
> | Linear (Ours) | **98.0$\pm$0.6** | **98.3$\pm$0.1** | **78.1$\pm$0.0** |
>
> **Q7: To facilitate a comprehensive and fair comparison, it is crucial to include an analysis of the model's speed in the paper.**
>
> We implemented a novel batch-wise parallel version of Dijkstra's algorithm to ensure fast training and evaluation of our model. The code has been provided in the supplementary material. We will include an analysis of the model's speed in comparison to all other methods in the camera ready paper.
>
> **Q8: To enhance the persuasiveness of the paper, it would be beneficial to include some more experiments, such as a) recent studies such as ISA [1] and SLASH [2] for the object discovery task, and b) property prediction results over the complex datasets such as CLEVRTEX, in the final version.**
>
> Code for ISA is currently unavailable, hence we refer to the table in Q2 for additional experiments with SLASH for object discovery on new datasets. We also show the updated results of property prediction with CLEVRTEX below:
>
> | Property | Position | Shape |
> |---|---|---|
> | Slot Attention | 47.5$\pm$19.6 | 30.6 |
> | EfficientMORL | 21.8$\pm$2.0 | 18.5 |
> | GENESIS-V2 | 79.8$\pm$8.0 | 35.2 |
> | SLATE | 62.0$\pm$8.0 | 30.5 |
> | SysBinder | 38.2$\pm$3.1 | 29.6 |
> | BO-QSA | 66.5$\pm$0.3 | 28.7 |
> | OC-Net | **80.7$\pm$2.4** | **36.1** |
>
> **Q9: The explanation regarding the mask transformation matrix (denoted as A) lacks clarity. It is crucial to provide additional details regarding the initialization and training of the matrix, going beyond the information provided in section A.4.**
>
> We will include additional description and further studies for the mask transformation matrix in the camera ready paper.
>
> [3a] Yang, Yafei, and Bo Yang. "Promising or elusive? unsupervised object segmentation from real-world single images.", NeurIPS 2022.
>
> [3b] Locatello, Francesco, et al., "Object-Centric Learning with Slot Attention", NeurIPS 2020.

---

> > ### Comment · Reviewer_T4Hn · 2023-08-15
> > **Official Comment by Reviewer T4Hn**
> >
> > The reviewer expresses appreciation for the authors' response, acknowledging that some concerns have been addressed through supplementary experiment results. However, the reviewer remains concerns about the scalability of the proposed method due to its simple concept, which, at the same time, diverges significantly from previous object-centric learning approaches. While the simplicity can be innovative, the reviewer emphasizes the need for rigorous validation to establish the method as a credible baseline framework for future research.
> >
> > In this context, the reviewer wish to note the importance of comprehensive details in the paper, regarding model architectures of both proposed and baseline models, as well as experimental settings and results. Specific concerns include the absence of detailed results for claims such as the clustering threshold ε and the impact of hyperparameters on performance. Additionally, the effectiveness of utilizing 1x1 convolutions needs empirical support, especially since it influences model robustness across diverse datasets.
> >
> > The reviewer also raises questions about the sensitivity of the proposed method to hyperparameters, like positional encoding. It seems that other positional encoding than the proposed linear encoding does not perform well, showing that it may affect performance across various datasets. The reviewer expresses concern regarding the potential performance decline of the proposed method when applied to different datasets with distinct model settings. The authors have not demonstrated how the method fares under these conditions, raising questions about its adaptability and generalization.
> >
> > Furthermore, the reviewer inquire about the proposed method's performance on object discovery tasks involving backgrounds. As claimed by the SLASH authors, understanding backgrounds is one of the important ability for the object-centric learning models. The reviewer also suggests the use of "FG-" metrics to clarify foreground-only benchmarks in the original paper.
> >
> > Lastly, the reviewer seeks more information on the real-world dataset experiments, including model results (does the model soley detect the target objects without any unexpected object discovery?) and dataset preprocessing details (where are the other parts including backgrounds in COCO?).
> >
> > While the reviewer acknowledges the paper's novelty and robust performance, the reviewer also assert that certain core details are lacking. The reviewers look forward to the authors' response addressing the raised concerns, as it would contribute to establishing a greater confidence in the validity and robustness of this paper.

---

> > > ### Author Response · Authors · 2023-08-17
> > >
> > > Thank you for your insightful and helpful follow-up feedback. We are happy to hear that we were able to address some of your concerns.
> > >
> > > **Q10: The reviewer wish to note the importance of comprehensive details in the paper, regarding model architectures of both proposed and baseline models, as well as experimental settings and results.**
> > >
> > > We used official setup and published code for the training of all baseline models (including BO-QSA [21], SLASH [5b], AST-Seg [3c]). Experimental settings were also followed from previous works when the same datasets are used. We will include full details for the new models and experiments in the camera-ready paper.
> > >
> > > **Q11: Specific concerns include the absence of detailed results for claims such as the clustering threshold $\epsilon$ and the impact of hyperparameters on performance.**
> > >
> > > The only hyperparameter of OC-Net that varies is the threshold $\epsilon$, and we have shown in Appendix A.3 that $\epsilon$ is able to take a wide range of values without affecting performance. This insensitivity to changes in $\epsilon$ holds for the new datasets as shown in the table below, where performance remains robust for a wide range of $\epsilon$ values:
> > >
> > > |  | Flowers | Birds | COCO |
> > > |---|---|---|---|
> > > | Metric | IoU-FG | IoU-FG | mIoU-FG |
> > > | OC-Net ($\epsilon = 0.7$) | 53.89$\pm$0.22 | 30.17$\pm$0.14 | 33.88$\pm$0.13 |
> > > | OC-Net ($\epsilon = 1.6$) | 54.41$\pm$0.22 | 31.77$\pm$0.15 | 35.57$\pm$0.14 |
> > > | OC-Net ($\epsilon = 2.3$) | **54.42 $\pm$0.23** | **33.50$\pm$0.17** | **35.58$\pm$0.15** |
> > >
> > > **Q12: The effectiveness of utilizing 1x1 convolutions needs empirical support, especially since it influences model robustness across diverse datasets.**
> > >
> > > As suggested, we conduct additional comparison with a new variant of OC-Net where the 1x1 convolutions are replaced with 5x5 convolutions. The mIoU results in the table below show that OC-Net with 1x1 convolutions produces more fine-grained segmentations especially for datasets with small objects:
> > >
> > > | | SVHN | IDRiD | CLEVRTEX | CLEVRTEX-OOD | Flowers | Birds | COCO |
> > > |---|---|---|---|---|---|---|---|
> > > | Metric | mIoU-FG | mIoU-FG | mIoU | mIoU | mIoU-FG | mIoU-FG | mIoU-FG |
> > > | OC-Net (5x5) | 42.1$\pm$0.2 | 19.1$\pm$0.1 | 30.3$\pm$0.7 | 31.2$\pm$0.2 | 53.6$\pm$0.2 | 32.5$\pm$0.2 | 27.5$\pm$0.1 |
> > > | OC-Net | **49.9$\pm$0.1** | **31.2$\pm$0.2** | **34.4$\pm$0.9** | **32.3$\pm$0.5** | **54.4$\pm$0.2** | **33.5$\pm$0.2** | **35.6$\pm$0.2** |
> > >
> > > **Q13: It seems that other positional encoding than the proposed linear encoding does not perform well, showing that it may affect performance across various datasets.**
> > >
> > > While using sinusoidal positional encoding affects property prediction results after training the extracted object representations with a gradient boosted tree, performance for object discovery remains robust to changes in positional encoding (PE) as seen from results in the table below on Multi-dSprites:
> > >
> > > | Metric | ARI-FG | Dice-FG | mIoU-FG |
> > > |---|---|---|---|
> > > | OC-Net (Sinusoidal PE) | 99.7$\pm$0.0 | 99.3$\pm$0.1 | 98.9$\pm$0.1 |
> > > | OC-Net (Linear PE) | **99.8$\pm$0.0** | **99.5$\pm$0.0** | **99.1$\pm$0.0** |
> > >
> > > **Q14: The reviewer inquire about the proposed method's performance on object discovery tasks involving backgrounds. As claimed by the SLASH authors, understanding backgrounds is one of the important ability for the object-centric learning models.**
> > >
> > > For rigorous evaluation of fine-grained object segmentation, we use foreground only (FG) metrics for the simulated (Multi-dSprites, Tetrominoes) and real-world datasets (SVHN, IDRiD, Flowers, Birds, COCO).
> > > We use foreground and background metrics for the complex textures datasets (CLEVRTEX, CLEVRTEX-OOD), where results in table 2(c) demonstrate that OC-Net is robust in segmenting complex-textured objects and backgrounds.
> > >
> > > **Q15: The reviewer also suggests the use of "FG-" metrics to clarify foreground-only benchmarks in the original paper.**
> > >
> > > We will use "FG-" metrics to clarify foreground-only benchmarks in the camera-ready paper.
> > >
> > > **Q16: Lastly, the reviewer seeks more information on the real-world dataset experiments, including model results (does the model soley detect the target objects without any unexpected object discovery?).**
> > >
> > > Following previous works, OC-Net fully segments each image before matching the ground-truth regions with the predicted regions for evaluation.
> > >
> > > **Q17: On dataset preprocessing details: where are the other parts including backgrounds in COCO?**
> > >
> > > We used the exact preprocessing steps as presented in [3d] and BO-QSA [21] for the Flowers, Birds and COCO datasets.
> > >
> > > [3c] Sauvalle and Fortelle. "Unsupervised multi-object segmentation using attention and soft-argmax.", WACV 2023.
> > >
> > > [3d] Yang, Yafei, and Bo Yang. "Promising or elusive? unsupervised object segmentation from real-world single images.", NeurIPS 2022.

---

> > > > ### Comment · Reviewer_T4Hn · 2023-08-18
> > > > **Official Comment by Reviewer T4Hn**
> > > >
> > > > Appreciate the authors' response and most of the concerns are resolved. The reviewer suggests that the authors provide a more detailed description of the circumstances, conduct a thorough review of the experimental results, and expand their empirical studies. It would be beneficial to extend these studies beyond the use of simple datasets, like Multi-d Sprites, to include more complex datasets.
> > > >
> > > > Maintaining the previous rating, the reviewer supports the acceptance of this paper. The reviewer anticipate that this paper will be shared with fellow researchers in the near future, and introduce a fresh and captivating path within our community.

---

> > > > > ### Author Response · Authors · 2023-08-18
> > > > >
> > > > > Thank you for your constructive feedback. We are happy to hear that we were able to address most of your concerns.

---

### Official Review · Reviewer_GCAa · 2023-07-05

**Soundness:** 2 fair
**Presentation:** 3 good
**Contribution:** 2 fair
**Rating:** 5
**Confidence:** 3

**Summary:**

The paper presents a method for learning pixel representations for multi-object segmentation. The method works by projecting pixels using $1 \times 1$ convolution into a feature space. The graph is defined on top of pixels using 8-connectivity. A modified Dijkstra's algorithm assigns pixels to objects based on a distance threshold, where the distance between pixels is measured as the Euclidean distance between normalised feature embeddings. Object representation is formed by averaging the pixel embeddings and adding a learnable mask projection. Two losses are used to learn the pixel embeddings, which aim to increase the diagonal and minimise the off-diagonal entries of the object-feature covariance matrix. The method is evaluated on a series of 2D sprite datasets, SVHN and IDRiD datasets and CLEVRTEX, where the method shows strong segmentation performance.

**Strengths:**


The presented formulation is quite simple. This leads to short training time and high-sample efficiency.

The method shows strong performance on benchmarks to the point of saturating or "solving" simpler cases.

The writing is clear.


**Weaknesses:**


The description of the method is lacking critical information and details.
 - In particular, it is not clear how final masks are obtained. Description of the method in Algorithm 1 suggests that the object sets $\mathcal{O}_c$ are not disjoint and may contain the same pixels as other objects (a consequence of revisiting all pixels starting from the empty set (Lines 11 and 24 of the algorithm)). How are the masks obtained in that case, e.g. for evaluation?
 - It is also not clear why the object discovery procedure converges, as it relies upon having learned an already appropriate pixel embedding to always guarantee that the condition in line 21 of the algorithm can be satisfied. What happens if that is not the case?

As there is no information exchange between pixels in the same object, it is not clear whether semantic or instance information can be learned and extracted. This is somewhat evidenced by the model not performing well on Tetrominoes in the case where the model is required to learn object shapes. As evaluation is limited to simpler or simulated datasets, it is not clear whether this would scale to complex real-world data. Given performance on simpler datasets, it would interesting to show the method working on Birds, COCO, and Flowers in comparison to prior work [21, A].

There are related works that explore image segmentation by modelling a graph on top of learned or hand-crafted features [B, C, D]. It would be beneficial to discuss and compare to such works, e.g. MaskCut method from [C].

Comparison to prior work is limited to older methods, whereas newer methods are not compared against. In particular, a comparison to [21, A, E] is needed given the strong performance on some of the same datasets. Furthermore, methods that were previously shown to obtain good results on CLEVRTEX are not included [E, F, G] in comparisons. Additionally, the object property prediction is limited to simple, nearly monochrome datasets whereas prior work [41] have considered more complex cases in CLEVRTEX already.


### References

[A] Seitzer et al. "Bridging the Gap to Real-World Object-Centric Learning"

[B] Wang et al. "Self-supervised Transformers for Unsupervised Object Discovery using Normalized Cut"

[C] Want et al. "Cut and learn for unsupervised object detection and instance segmentation"

[D] Shi and Malik "Normalized cuts and image segmentation"

[E] Sauvalle and Fortelle "Unsupervised multi-object segmentation using attention and soft-argmax"

[F] Jiang and Ahn "Generative Neurosymbolic Machines"

[G] Monnier et al. "Unsupervised layered image decomposition into object prototypes"


**Questions:**


Is it correct to say that the LayerNorm is applied _without_ learned affine transformation as outputs of this layer (or functions thereof) are not included in the loss calculation? If it is, how is it trained as the membership of pixel to object set does not appear to be differentiable?

How is the Figure 2 visualisation obtained? It seems that the object discovery procedure requires pixels to be connected, however, there are several "floating" pixels coloured the same colour as the objects.

On L161, $\epsilon$ is described as being set such that the normalised similarity should be 50%. How are the similarities normalised?


**Limitations:**


There seems to be an unstated assumption that objects need to be up to a certain size. I.e. the threshold implicitly defines a maximum distance between the sampled initial pixel and the furthest pixel that can be included. For example, considering an image of solid color, the resulting pixels should have the same features (close to zero due to LayerNorm) save for the added positional embedding. Thus, only pixels $\epsilon$ from the starting pixel will be added to the object set $\mathcal{O}_c$.

It also seems that the assignment of the pixels to a particular object might be heavily dependent on the sampled initial pixel. That is, a pixel j might be included if the initial sampled pixel i is close to it, but not if it is further away as it cannot satisfy the condition on Line 21 of Algorithm 1 due to too many hops. It will then become a part of a different object. Or conversely, pixels of two different objects from a boundary would be included in the same object if one was sampled as initial pixels. Training would further enforce this. Considering the situation above, if the initial sampled pixel was in the corner, would this not potentially create several components?

Similarly, could it be that the model cannot handle an object split into two regions by another object, such as e.g. a dog behind a fence post?

---

> ### Author Rebuttal · Authors · 2023-08-08
>
> Thank you for your detailed review and for the extensive pointers to related works. We provide some responses as follows:
>
> **Q1: Algorithm 1 suggests that the object sets are not disjoint and may contain the same pixels as other objects. How are the masks obtained in that case, e.g. for evaluation?**
>
> If a pixel is assigned to multiple objects, we assign it to the mask of the first object in that list and ignore its membership in other objects.
>
> **Q2: It is not clear why the object discovery procedure converges, as it relies upon having learned an already appropriate pixel embedding to always guarantee that the condition in line 21 of the algorithm can be satisfied. What happens if that is not the case?**
>
> There is always at least one pixel (the starting pixel $\mathbf{p}_i$) that satisfies the condition in line 21, thus guaranteeing that the algorithm will terminate.
>
> **Q3: As there is no information exchange between pixels in the same object, it is not clear whether semantic information can be learned and extracted.**
>
> Relational information between pixels in the same object are included in object representations in the form of positional encodings and the processed object mask (Eqn. 2). This enables our model to outperform other baselines in downstream property prediction, demonstrating that semantic information can be extracted.
>
> **Q4: Given performance on simpler datasets, it would interesting to show the method working on Birds, COCO, and Flowers**
>
> We have conducted additional experiments for further evaluation on Flowers, Birds, COCO as suggested. We have also added two more recent baselines BO-QSA [21] and SLASH [5b] for comparison. The results in the table below shows that OC-Net is robust and remains the top performer:
>
> |  | Flowers | Flowers | Birds | Birds | COCO | COCO |
> |---|---|---|---|---|---|---|
> | Metric | Dice | IoU | Dice| IoU | mDice| mIoU |
> | SLASH | 39.9$\pm$1.5 | 26.0$\pm$1.2 | 43.5$\pm$1.1 | 28.2$\pm$1.1 | 22.3$\pm$1.1 | 13.4$\pm$1.1 |
> | BO-QSA | 65.8$\pm$1.9 | 51.7$\pm$1.9 | 44.6$\pm$1.7 | 30.3$\pm$1.5 | 34.9$\pm$1.1 | 23.6$\pm$0.9 |
> | OC-Net | **67.2$\pm$0.2** | **54.4$\pm$0.2** | **47.8$\pm$0.2** | **33.5$\pm$0.2** | **48.2$\pm$0.2** | **35.6$\pm$0.2** |
>
> The objects discovered for the various methods on sample images can be viewed in the PDF attached in our general response to all reviewers. We see that OC-Net is able to segment out the objects in a fine-grained manner.
>
> **Q5: There are related works that explore image segmentation by modelling a graph on top of learned or hand-crafted features [B, C, D]. It would be beneficial to discuss and compare to such works.**
>
> DINOSAUR [A], TokenCut [B], MaskCut [C] belong to a class of methods that rely on the DINO model [2a] which was pre-trained on the ImageNet dataset. For fair comparison, we excluded methods that rely on pre-trained models and trained all methods from random initialization. Nonetheless, we will include a section to compare these works in the updated paper as suggested.
>
> We have carried out additional experiments for comparison with Ncut [D] as suggested. The mIoU results in our general response (3) show that OC-Net outperforms other graph-based methods.
>
> **Q6: A comparison to [21,A,E] is needed given the strong performance on some of the same datasets. Furthermore, methods that were previously shown to obtain good results on CLEVRTEX are not included [E,F,G] in comparisons.**
>
> We conduct additional comparison with most recently published BO-QSA [21] which attain scores that outperform or are comparable with AST-Seg [E], GNM [F], DTI [G]. The mIoU results in our general response (4) shows that OC-Net is superior.
>
> **Q7: The object property prediction is limited to simple, nearly monochrome datasets whereas prior work have considered more complex cases in CLEVRTEX already.**
>
> We conducted additional experiments to further evaluate the object property prediction on CLEVRTEX as suggested. Results are in our general response (2).
>
> **Q8: Is it correct to say that the LayerNorm is applied without learned affine transformation as outputs of this layer (or functions thereof) are not included in the loss calculation?**
>
> Yes, LayerNorm is applied without affine transformation. Membership of pixel to object set is used to control the flow of gradients such that gradients only flow towards the pixel embeddings which belong to each object.
>
> **Q9: It seems that the object discovery procedure requires pixels to be connected, however, there are several "floating" pixels coloured the same colour as the objects.**
>
> As clarified in Q1, pixels may be assigned to multiple objects and during evaluation, we assign such pixels to the mask of the first object. This may result in some floating pixels.
>
> **Q10: $\epsilon$ is described as being set such that the normalised similarity should be 50\%. How are the similarities normalised?**
>
> We normalise all similarity values to be between $0$ and $1$ by taking the negative exponent. Details have been provided in Appendix A.3.
>
> **Q11: There seems to be an unstated assumption that objects need to be up to a certain size. It also seems that the assignment of the pixels to a particular object might be heavily dependent on the sampled initial pixel.**
>
> Objects can include any pixel as long as there exists a shortest path distance that is less than $\epsilon$ from that pixel to the sampled pixel. We have found that our trained models are robust to size and sampling sequence as demonstrated by $\epsilon$ being able to take a wide range of values without affecting performance (see Appendix A.3).
>
> **Q12: Could it be that the model cannot handle an object split into two regions by another object?**
>
> Yes, as stated in the limitations on occlusion, if an object is split into two disjoint regions by another object, the model would consider them as separate objects and output three object masks.
>
> [2a] Caron et al. "Emerging properties in self-supervised vision transformers", ICCV 2021.

---

> > ### Comment · Reviewer_GCAa · 2023-08-14
> >
> > I thank the authors for their response. I believe some of my concerns have been addressed.
> >
> > It is particularly encouraging to see the results on the more varied real-world datasets, which strengthen the paper's message greatly.
> >
> > I still have some concerns regarding the presented algorithmic procedure. Mainly the necessity to ignore potential multiple object assignments, the potential randomness resulting from the uniform sampling of the initial pixel and other limitations needs discussion and more focus in the paper. Details such as the answer to Q1 are critical to understanding the method, so I would encourage incorporating them into the methods section.
> >
> > Also statements such as "BO-QSA [21] which attain scores that outperform or are comparable with AST-Seg [E], GNM [F], DTI [G]" need to be supported with experimental results, given that such comparisons were not presented in [21] (except for [G]).
> >
> > Finally, it would be good to include (spate/time permitting) the descriptions of the baseline models used to make the new comparisons in the paper.
> >
> > For example, is BO-QSA a transformer or mixture decoder-based model? The results are in Tab 11. (additional PDF) seem to be better than the mixture version from [21] but much lower than the transformer version. Was the model also altered as SLATE/SysBinder (Appendix C) to have 1x1 inputs?
> >
> > I will see if other reviewers have any new questions that develop into a discussion but will be updating my score to reflect the new results and clarifications presented in the rebuttal afterward.

---

> > > ### Author Response · Authors · 2023-08-17
> > >
> > > Thank you for your constructive follow-up feedback. We are happy to hear that we were able to address some of your concerns.
> > >
> > > **Q13: I still have some concerns regarding the presented algorithmic procedure. Mainly the necessity to ignore potential multiple object assignments, the potential randomness resulting from the uniform sampling of the initial pixel**
> > >
> > > During the evaluation of 2D images where each pixel can only belong to one ground-truth object, we assign pixels that have been assigned to multiple objects to the mask of the first object. When pixels could belong to multiple ground-truth objects, we retain the multiple object assignments.
> > >
> > > The sampling-based algorithm gives OC-Net the key benefit of being able to segment any image without the need to fix the number of objects beforehand, as required in most state-of-the-art methods.
> > >
> > > **Q14: Details such as the answer to Q1 are critical to understanding the method, so I would encourage incorporating them into the methods section.**
> > >
> > > We will include these details in the updated paper.
> > >
> > > **Q15: Statements such as 'BO-QSA [21] which attain scores that outperform or are comparable with AST-Seg [E], GNM [F], DTI [G]' need to be supported with experimental results, given that such comparisons were not presented in [21] (except for [G]).**
> > >
> > > Since AST-Seg [E] presented scores which outperform GNM [F] and DTI [G], we conduct additional comparison with AST-Seg as suggested. We train all models end-to-end for fair comparison. The table below shows the results on CLEVRTEX and CLEVRTEX-OOD with $64\times 64$ images, we see that OC-Net remains competitive and is robust especially for fine-grained segmentation of foreground objects (mIoU-FG):
> > >
> > > |  | CLEVRTEX | CLEVRTEX | CLEVRTEX-OOD | CLEVRTEX-OOD |
> > > |---|---|---|---|---|
> > > | Metric | mIoU-FG | mIoU | mIoU-FG | mIoU |
> > > | BO-QSA | 32.72 $\pm$1.65 | 34.70 $\pm$1.33 | 31.91 $\pm$1.40 | 33.92 $\pm$1.13 |
> > > | AST-Seg-B3-BT | 31.77 $\pm$1.72 | **38.63 $\pm$1.81** | 25.93 $\pm$1.87 | 32.41 $\pm$2.17 |
> > > | OC-Net | **35.73 $\pm$0.07** | **37.49 $\pm$0.07** | **33.93 $\pm$0.13** | **34.98 $\pm$0.14** |
> > >
> > > **Q16: It would be good to include (space/time permitting) the descriptions of the baseline models used to make the new comparisons in the paper. For example, is BO-QSA a transformer or mixture decoder-based model? Was the model also altered as SLATE/SysBinder (Appendix C) to have 1x1 inputs?**
> > >
> > > We will include full details for the new experiments in the updated paper. We use the unaltered transformer version from [21] with $64$ latent size.
> > > Results in Table 11 were computed only with respect to the foreground for the Flowers and Birds datasets to evaluate fine-grained segmentation.
> > > We present the mIoU and mDice scores for foreground and background in the table below, which show that OC-Net remains robust:
> > >
> > > |  | Flowers | Flowers | Birds | Birds |
> > > |---|---|---|---|---|
> > > | Metric | mDice | mIoU | mDice| mIoU |
> > > | SLASH | 62.18 $\pm$1.04 | 50.15 $\pm$1.99 | 67.20$\pm$1.45 | 55.93$\pm$1.32 |
> > > | BO-QSA | 75.11 $\pm$0.47 | 63.01 $\pm$0.37 | 67.79$\pm$0.76 | 56.95$\pm$0.23 |
> > > | OC-Net | **75.47 $\pm$0.09** | **63.84 $\pm$0.10** | **68.09$\pm$0.08** | **57.32$\pm$0.08** |

---

> > > > ### Comment · Reviewer_GCAa · 2023-08-18
> > > >
> > > > Based on added experimental evidence and provided answers, I have updated my rating as previously indicated.

---

> > > > > ### Author Response · Authors · 2023-08-18
> > > > >
> > > > > Thank you for updating your rating. We are happy to hear that we were able to address most of your concerns.

---

### Official Review · Reviewer_TJh8 · 2023-07-07

**Soundness:** 3 good
**Presentation:** 4 excellent
**Contribution:** 3 good
**Rating:** 8
**Confidence:** 2

**Summary:**

The paper discusses a novel method for learning object-centric representations from images, the new approach uses feature connectivity to group together pixels likely to be part of the same object and employs two object-centric regularization terms for refining the representations.

The method has been tested on various types of images and has significant improvements in object discovery quality, sample efficiency, and generalizability compared to prior work. It has the potential to improve many downstream tasks in predicting object properties. But it also has limitations to handle objects with complex part-whole hierarchies and object occlusions.

**Strengths:**

1. Contributed a novel method for learning object-centric representations from images, the Object Discovery process and Object-Centric Regularization are carefully designed for object-centric learning.

2. Comprehensive mathematical proof and reasoning for the proposed embeddings and regularization terms.

3. Comprehensive comparison between the proposed methods and prior work, and on various datasets.

4. Promising results on sample efficiency, accuracy, and generalizability compared to prior work.

5. Paper writing is clear and succinct.

**Weaknesses:**

1. Lack of example on incorporating the method to an actual downstream task, and showing the advantage of the method using the proposed terms.

2. Lack of results on more challenging datasets with complex scenes, object types, object shapes, and object occlusions.

3. Designed for object-centric, but as mentioned in the limitations, it has limitations for objects with complex part-whole hierarchies.

**Questions:**

How does the proposed object entanglement compare to the attention mechanism?

**Limitations:**

The authors mentioned the method has limitations to handle objects with complex part-whole hierarchies and object occlusions. It would still be interesting to see how the method works on more realistic images, such as a picture of a complex kitchen scene, with part-whole complex cabinetry and a human holding objects and having big object occlusions.

---

> ### Author Rebuttal · Authors · 2023-08-08
>
> Thank you for your detailed review and for the encouraging comments. We provide some responses to the questions as follows:
>
> **Q1: Lack of results on more challenging datasets with complex scenes, object types, object shapes, and object occlusions.**
>
> We have conducted additional experiments on the Birds, COCO and Flowers datasets with more complex scenes as suggested. We have also added two more recent baselines BO-QSA [21] and SLASH [1a] for comparison. The table below shows the results:
>
> |  | Flowers | Flowers | Birds | Birds | COCO | COCO |
> |---|---|---|---|---|---|---|
> | Metric | Dice | IoU | Dice| IoU | mDice| mIoU |
> | SLASH | 39.91 $\pm$1.48 | 25.98 $\pm$1.15 | 43.45$\pm$1.10 | 28.24$\pm$1.08 | 22.34$\pm$1.10 | 13.37$\pm$1.07 |
> | BO-QSA | 65.77 $\pm$1.90 | 51.70 $\pm$1.93 | 44.61$\pm$1.65 | 30.29$\pm$1.50 | 34.90$\pm$1.14 | 23.57$\pm$0.94 |
> | OC-Net | **67.21 $\pm$0.21** | **54.42 $\pm$0.23** | **47.80$\pm$0.19** | **33.50$\pm$0.17** | **48.18$\pm$0.17** | **35.58$\pm$0.15** |
>
> The objects discovered for the various methods on sample images can be viewed in the PDF attached in our general response to all reviewers.
>
> **Q2: The authors mentioned the method has limitations to handle objects with complex part-whole hierarchies and object occlusions. It would still be interesting to see how the method works on more realistic images, such as a picture of a complex kitchen scene, with part-whole complex cabinetry and a human holding objects and having big object occlusions.**
>
> We will include some of these samples in the camera ready paper.
>
> **Q3: How does the proposed object entanglement compare to the attention mechanism?**
>
> The Slot Attention baseline [1b] used in the paper relies on an adapted version of the attention mechanism to perform object-centric learning. This attention mechanism does not consider feature connectivity whereas our method does, enabling better fine-grained object discovery and sample efficiency.
>
> [1a] Kim, Jinwoo, et al., "Shepherding Slots to Objects: Towards Stable and Robust Object-Centric Learning", CVPR 2023.
>
> [1b] Locatello, Francesco, et al., "Object-Centric Learning with Slot Attention", NeurIPS 2020.

---

> > ### Comment · Reviewer_TJh8 · 2023-08-21
> >
> > Thank you for providing further experimental results and addressing my questions.
> >
> > All my concerns have been satisfactorily addressed, and I continue my rating as strong accept. Tackling multiple object learning, especially in complex real-world scenarios is challenging. The paper introduces a novel and promising approach to multiple object representation learning, making a valuable contribution to this field.

---

### Author Rebuttal · Authors · 2023-08-09

We would like to thank the reviewers for the helpful feedback and insightful comments.

We are encouraged that our contribution of an unsupervised object discovery network which leverages on feature connectivity and designed object-centric regularization terms is described as "*novel*" (TJh8, T4Hn), "*innovative*" (2W4b), "*simple*" (GCAa, 2W4b) and "*yet it proves to be highly efficacious.*" (2W4b).

All reviewers highlight the "*sample efficiency*" (TJh8, T4Hn, GCAa) and "*applicability of the proposed design even when the scale of the dataset diminishes.*" (2W4b). We are also heartened that our efforts to design a solution based on theoretical grounds has been described as "*comprehensive*" (TJh8), "*enhancing our understanding of the proposed method's performance and its capacity to generalize effectively in downstream tasks*" (T4Hn). Finally, reviewers comment that our experiments "*on six diverse datasets*" (TJh8) are "*comprehensive*" (Tjh8), "*encouraging*" (2W4b) and "*shows strong performance on benchmarks to the point of saturating or 'solving' simpler cases*" (GCAa).

We further thank reviewers for the very constructive feedback which help us refine our work. In response to this feedback, we have conducted new experiments which we summarize below:

**1. Application of OC-Net to more complex scenes**

We have conducted new experiments for further evaluation on additional Flowers, Birds, COCO [5a] datasets which contain objects that have more diverse colors and shapes. We have also added two more recent baselines BO-QSA [21] and SLASH [5b] for comparison as suggested. The results in the table below shows that OC-Net is robust and remains the top performer:

|  | Flowers | Flowers | Birds | Birds | COCO | COCO |
|---|---|---|---|---|---|---|
| Metric | Dice | IoU | Dice| IoU | mDice| mIoU |
| SLASH | 39.91$\pm$1.48 | 25.98$\pm$1.15 | 43.45$\pm$1.10 | 28.24$\pm$1.08 | 22.34$\pm$1.10 | 13.37$\pm$1.07 |
| BO-QSA | 65.77$\pm$1.90 | 51.70$\pm$1.93 | 44.61$\pm$1.65 | 30.29$\pm$1.50 | 34.90$\pm$1.14 | 23.57$\pm$0.94 |
| OC-Net | **67.21$\pm$0.21** | **54.42$\pm$0.23** | **47.80$\pm$0.19** | **33.50$\pm$0.17** | **48.18$\pm$0.17** | **35.58$\pm$0.15** |

The objects discovered for the various methods on sample images can be viewed in the attached PDF. We see that OC-Net is able to segment out the objects in a fine-grained manner.

**2. Object property prediction evaluation on CLEVRTEX**

We conducted additional experiments to further evaluate the object property prediction on CLEVRTEX as suggested. The table below shows the results:

| Property | Position | Shape |
|---|---|---|
| Slot Attention | 47.5$\pm$19.6 | 30.6 |
| EfficientMORL | 21.8$\pm$2.0 | 18.5 |
| GENESIS-V2 | 79.8$\pm$8.0 | 35.2 |
| SLATE | 62.0$\pm$8.0 | 30.5 |
| SysBinder | 38.2$\pm$3.1 | 29.6 |
| BO-QSA | 66.5$\pm$0.3 | 28.7 |
| OC-Net | **80.7$\pm$2.4** | **36.1** |

**3. Additional comparison with new unsupervised segmentation algorithm**

We have carried out additional experiments for comparison with Ncut [5c] as suggested. The mIoU results in the table below show that OC-Net significantly outperforms other graph-based methods:

| Method | Multi-dSprites | Tetrominoes | SVHN | IDRiD | CLEVRTEX | CLEVRTEX-OOD |
|---|---|---|---|---|---|---|
| Felzenszwalb | 95.0$\pm$0.0 | 96.9$\pm$0.0 | 39.8$\pm$0.0 | 15.4$\pm$0.0 | 26.8$\pm$0.0 | 23.4$\pm$0.0 |
| Ncut | 58.9$\pm$0.0 | 57.4$\pm$0.0 | 32.2$\pm$0.0 | 4.3$\pm$0.0 | 22.9$\pm$0.0 | 18.6$\pm$0.0 |
| OC-Net | **99.1$\pm$0.0** | **100.0$\pm$0.0** | **49.9$\pm$0.1** | **31.2$\pm$0.2** | **34.4$\pm$0.9** | **32.3$\pm$0.5** |

**4. Additional comparison with new state-of-the-art model**

We conduct additional comparison with the most recently published BO-QSA model [21]. The mIoU results in the table below shows that OC-Net is superior:

| Method | Multi-dSprites | Tetrominoes | SVHN | IDRiD | CLEVRTEX | CLEVRTEX-OOD |
|---|---|---|---|---|---|---|
| BO-QSA | 88.0$\pm$1.2 | 25.8$\pm$1.2 | 48.3$\pm$1.3 | 4.5$\pm$1.7 | 31.9$\pm$1.7 | 31.2$\pm$1.0 |
| OC-Net | **99.1$\pm$0.0** | **100.0$\pm$0.0** | **49.9$\pm$0.1** | **31.2$\pm$0.2** | **34.4$\pm$0.9** | **32.3$\pm$0.5** |

[5a] Yang, Yafei, and Bo Yang. "Promising or elusive? unsupervised object segmentation from real-world single images.", NeurIPS 2022.

[5b] Kim, Jinwoo, et al., "Shepherding Slots to Objects: Towards Stable and Robust Object-Centric Learning", CVPR 2023.

[5c] Shi, Jianbo, and Jitendra Malik. "Normalized cuts and image segmentation." IEEE Transactions on pattern analysis and machine intelligence, 2000.

---

### Decision · Program_Chairs · 2023-09-21

**Decision:**

Accept (spotlight)

**Comment:**

This submission has received four reviews that recommend acceptance. The authors provided extensive answers to the reviewers in several rounds of discussion and provided new evidence. All reviewers wrote that their questions has been answered and resolved to the extend possible in this format and recommend acceptance.

Since all reviewers appreciate the approach as innovative and novel and find the experiments supportive of the claims we recommend a spotlight presentation for this paper. This paper extends on object centric learning with an approach to connect features and the results surpass prior work on relevant datasets. This will be of interest to the object-centric researchers but appears to be a good showcase to present to the wider NeurIPS community.